# Specialized Metabolites Accumulation Pattern in Buckwheat Is Strongly Influenced by Accession Choice and Co-Existing Weeds

**DOI:** 10.3390/plants12132401

**Published:** 2023-06-21

**Authors:** Yedra Vieites-Álvarez, Paz Otero, David López-González, Miguel Angel Prieto, Jesus Simal-Gandara, Manuel J. Reigosa, M. Iftikhar Hussain, Adela M. Sánchez-Moreiras

**Affiliations:** 1Departamento de Bioloxía Vexetal e Ciencia do Solo, Facultade de Bioloxía, Campus Lagoas-Marcosende s/n, Universidade de Vigo, 36310 Vigo, Spain; yedra.vieites.alvarez@uvigo.es (Y.V.-Á.); davidlopez@uvigo.es (D.L.-G.); mreigosa@uvigo.es (M.J.R.); iftikhar@uvigo.es (M.I.H.); 2Nutrition and Bromatology Group, Department of Analytical and Food Chemistry, Faculty of Food Science and Technology, Universidade de Vigo, Ourense Campus, 32004 Ourense, Spain; potero@uvigo.es (P.O.); mprieto@uvigo.es (M.A.P.); jsimal@uvigo.es (J.S.-G.); 3Instituto de Agroecoloxía e Alimentación (IAA), Universidade de Vigo, Campus Auga, 32004 Ourense, Spain

**Keywords:** buckwheat, polyphenols, sustainable weed control, allelopathy, competition

## Abstract

Screening suitable allelopathic crops and crop genotypes that are competitive with weeds can be a sustainable weed control strategy to reduce the massive use of herbicides. In this study, three accessions of common buckwheat *Fagopyrum esculentum* Moench. (Gema, Kora, and Eva) and one of Tartary buckwheat *Fagopyrum tataricum* Gaertn. (PI481671) were screened against the germination and growth of the herbicide-resistant weeds *Lolium rigidum* Gaud. and *Portulaca oleracea* L. The chemical profile of the four buckwheat accessions was characterised in their shoots, roots, and root exudates in order to know more about their ability to sustainably manage weeds and the relation of this ability with the polyphenol accumulation and exudation from buckwheat plants. Our results show that different buckwheat genotypes may have different capacities to produce and exude several types of specialized metabolites, which lead to a wide range of allelopathic and defence functions in the agroecosystem to sustainably manage the growing weeds in their vicinity. The ability of the different buckwheat accessions to suppress weeds was accession-dependent without differences between species, as the common (Eva, Gema, and Kora) and Tartary (PI481671) accessions did not show any species-dependent pattern in their ability to control the germination and growth of the target weeds. Finally, Gema appeared to be the most promising accession to be evaluated in organic farming due to its capacity to sustainably control target weeds while stimulating the root growth of buckwheat plants.

## 1. Introduction

Pests and weeds are probably the primary biotic limitation that farmers face when attempting to increase the yield of their crops. Therefore, research, development, and the use of chemical products exclusively produced for the control of weeds/pests in the field has increasingly grown in the last decades. However, these synthetic chemical products, which have been extensively applied in the field, are dangerous to terrestrial environment and to human health [1], and increase weed resistance [2]. Several synthetic herbicides, including glyphosate, exhibit significant potential for soil adsorption and cannot move around freely in the environment. The effects of herbicides on human health depend on the concentration, length, and frequency of exposure, and often lead to cytotoxic and DNA damage and carcinogenicity [3]. Reducing their use is an increasing necessity in order for more sustainable food production.

There are several recent reports that highlight the development of resistance in different *Lolium* sp., including *Lolium rigidum* Gaudin, the species used in this study, which is likely the cause of the significant loss in cereal crop yield in Mediterranean countries and Australia [4,5]. In the US, populations of Italian ryegrass (*L. perenne* ssp. *multiflorum* (Lam.) Husn.) have also been found to be resistant to glufosinate [6]. As a result of the quick evolution of resistance in *L. rigidum*, comprehensive weed management measures, including crop allelopathic varieties, are required to slow down this rapid evolution and sustainable control. In several European countries, including Spain and Portugal, resistance to acetyl-CoA carboxylase (ACCase), acetolactate synthase (ASL), photosystem II (PSII), 5-enolpyruvylshikimate-3-phosphate synthase (EPSPS), glutamine synthase, very long-chain fatty acid (VLCFA) synthesis, and protoporphyrinogen oxidase (PPO)-inhibiting herbicides has been reported in *L. rigidum*, which is mainly associated with winter cereals, vineyards, and orchards [7,8]. The other species used in this study, *Portulaca oleracea* L., commonly known as purslane, is a weed belonging to the Portulacaceae family that has been ranked ninth out of the world’s worst weeds, and has been recorded in 45 crops in 81 countries [9]. According to the International Herbicide-Resistant Weed Database, *P. oleracea* has been already found to develop resistance to Group 5 (Legacy C1 C2) herbicides, which are referred to as PSII inhibitors—Serine 264 Binders. These specific biotypes are resistant to the atrazine and linuron herbicides, and may also be cross-resistant to other members of the Group 5 (Legacy C1 C2) family. Other researchers have also reported *P. oleracea* as a resistant weed to linuron [10]. This is a common phenomenon in many weeds due to the massive use of synthetic herbicides.

Several integrated weed management strategies can be employed to limit herbicide use, such as harrowing or hoeing, flames or hot water, allelopathic cover crops, smother crops, and green mulching, which are crucial for controlling the weed seed population in the soil for green economy, plant biodiversity, and environmental sustainability [11]. The circular economy, promoted by the European Commission, defends the reuse of various types of organic biomass and organic waste, thus transforming waste management into economic opportunities [12,13]. In this context, screening suitable allelopathic crops and crop genotypes that are competitive with weeds can be a secure and environmentally responsible weed control strategy [14]. By releasing specialized metabolites into the surrounding environment that may act as phytotoxins to suppress weeds, they fall in the framework of a process known as allelopathy and provide themselves a competitive edge [15,16]. By encouraging diversification of the agricultural system and reducing the reliance on herbicides, this approach can also reduce the risk of the development of herbicide resistance in monocot and dicot weeds populations. Weed management must protect environmental quality and human health, and allelochemicals released from live crops and crop residues can be used in this way to manage weeds and improve crop performance [17].

The *Fagopyrum* genus is a dicotyledonous pseudo-cereal crop of the family Polygonaceae. It is comprised of both perennial and annual species with diploid (2n = 2x = 16) and tetraploid (2n = 4x = 32) chromosome numbers with a haploid genome size of  −1.2 Gb. Out of the 34 reported species to date, this genus is recognized primarily by two cultivated species, common buckwheat (*F. esculentum* Moench.) and Tartary buckwheat (*F. tataricum* (L.) Gaertn.), along with the wild species *F. cymosum* (Trev.) Meisn. [18]. Buckwheat, which is known for its high nutritional value and bioactive components, is cultivated mainly for the production of food and pharmacological products for humans, and is increasingly considered as a promising emergent crop, as it can be an important source of proteins with well-balanced amino acid composition, dietary fibre, and phenolic substances [19]. Buckwheat is rich in phenolic acids and flavonoids, and exhibits antioxidative properties [20,21,22]. Besides well-known antioxidative effects, it has recently been suggested that buckwheat, as a source of the flavonoid quercetin, may prevent health problems in patients with diabetes [19].

Regarding its allelochemical potential, Kumar et al. [23] demonstrated that buckwheat residues in the soil were able to reduce powell amaranth, shepherd’s purse, and corn chamomile growth. There are several research and review manuscripts published advocating that buckwheat tissues possess abundant specialized metabolites, especially phenolic acids and flavonoids, with a significant allelopathic potential [24,25,26] that could be responsible for the weed-suppressive activity of buckwheat [23,27,28]. In addition, it has been reported that the suppression of some weeds is caused by the light competition of rapidly growing buckwheat plants [29]. However, previous studies have shown that common buckwheat, thanks to allelochemicals, can also significantly reduce the biomass of certain weed species such as *Thlaspi arvense* L., *Cirsium arvense* (L.) Scop., and *Plantago lanceolata* L. [30].

Phenolic compounds, abundant metabolites in buckwheat [31,32], are well known for their allelochemical properties. Phenolic acids such as cinnamic, dihydroxybenzoic, ferulic, *p*-coumaric, phthalic, syringic, *m*-toluic, or protocatechuic have been related to the allelochemical-induced growth inhibition of several weed species [33]. These compounds can be found in different parts of the buckwheat plant, such as the leaves, stems, and roots [34]. When buckwheat residues, such as decomposed plant material or root exudates, are present in the soil, the phenolic acids released can exert inhibitory effects on weed germination, growth, and development [23,29]. The specific mechanism through which phenolic acids inhibit weeds is not fully understood, but it is believed to involve interference with weed seed germination, root elongation, and nutrient uptake. The research has demonstrated the allelopathic effects of buckwheat on various weed species. For example, studies have shown that buckwheat residues can inhibit the germination and growth of weeds such as common lambsquarters (*Chenopodium album*), pigweed (*Amaranthus* spp.), and barnyard grass (*Echinochloa crus-galli*) [23,34]. It is important to note that the allelopathic effects can vary depending on factors such as the buckwheat cultivar and the weed species in question.

Both competition and allelopathy, as mechanisms of plant interference, have been well documented under controlled conditions [35]. The combined effects of allelopathy and crop competition determine the total weed-suppressive potential of a given variety, and research groups worldwide have been working to improve both traits simultaneously in order to achieve maximum gains in weed suppression [36,37], particularly in cereal crops. The root length of ryegrass, mustard, and lettuce was significantly reduced when their seeds co-germinated with buckwheat [30]. Although more recent studies are mostly focused on the allelopathic action of common buckwheat on weeds, Tartary buckwheat has also been studied for years regarding its potential for weed management [38].

However, little is known about the direct relation among allelochemicals exudation or accumulation and weed control by common or Tartary buckwheat. Therefore, the objective of this research was to examine the impact of three accessions of common buckwheat *Fagopyrum esculentum* Moench. (Gema, Kora, and Eva), and one accession of Tartary buckwheat, *Fagopyrum tataricum* (L.) Gaertn. (PI481671), on monocot (*Lolium rigidum* Gaud.) and dicot (*Portulaca oleracea* L.) weeds under laboratory-based germination and seedling growth bioassays. The potential of allelopathic compounds was carried out through identifying and quantifying a broad polyphenols profile (phenolic acids and flavonoids) in the roots, shoots, and root exudates of the different buckwheat accessions to compare the abundance of these allelochemicals with their weed suppressive capacity. The specific objectives of this study include the elucidation of the allelopathic potential of selected buckwheat accessions and their possible use to promote plant-based herbicides and their use in sustainable weed management strategies.

## 2. Results

### 2.1. Germination and Seedling Growth

Our results show that the germination rate of both weed species, *L. rigidum* and *P. oleracea*, was significantly decreased following being co-cultured with the four tested buckwheat accessions (Gema, Kora, Eva, and PI481671), which showed the strong capacity to control weed germination following a 7-day co-culture and it was species-specific (Table 1). Only *P. oleracea* could maintain its germination unaltered in the presence of the Eva accession. Although different buckwheat accessions behaved differently, they induced significant inhibition of germination when compared with the control, with Gema (19%) and PI481671 (35%) being the most inhibitory accessions for *L. rigidum* followed by Eva (50%) and Kora (54%). On the contrary, Kora (41%) was the most inhibitory accession for *P. oleracea* (41%), followed by PI481671 (55%) and Gema (59%).

Regarding the impact of the presence of buckwheat accessions on weed growth and development, we obtained a species dependent behaviour again, with *L. rigidum* as the most sensitive weed species to buckwheat co-cultivation (Table 1). The most inhibited parameter was the fresh plant weight for *L. rigidum* in the presence of Kora (73%), PI481761 (68%), and Eva (52%), while the most stimulated parameter was the shoot length of *P. oleracea* in the presence of PI481761 (186%), Gema (158%) and Eva (146%). *L. rigidum* was especially sensitive to Eva, as shoot length significantly increased (149%) while reducing fresh plant weight (52%), which resulted in longer but weaker plants. Something similar was found for *P. oleracea* in front of Gema accession, which significantly increased both, shoot (158%) and root (166%) length while maintaining unaltered the total weight of the plant (107%).

Regarding the germination and growth parameters of the different buckwheat accessions in front of the two tested weeds (Table 2), our results show that the shoot and root lengths of Eva were significantly inhibited in front of *P. oleracea,* as well as the root length in front of *L. rigidum*. In fact, Eva was the only accession where the shoot and root length were statistically inhibited in the presence of the weeds, while Gema and PI481671 also showed significant stimulation of root length when co-cultured with *P. oleracea*. PI481671 showed a stimulation of leaf weight in front of *P. oleracea* when compared with the buckwheat plants growing alone. Finally, the root weight was only inhibited in Kora in the presence of the monocot weed *L. rigidum*. From the parameters measured in the different accessions, we could see also that common buckwheat accessions (Eva, Gema, and Kora) grew much more than Tartary buckwheat (PI481671), which showed shoot length values that were two times lower than common buckwheat and root lengths values that were even three times lower than Eva and Kora and two times lower than Gema. These differences were even stronger in the case of leaf weight, where Tartary buckwheat values were ten times under common buckwheat accessions.

### 2.2. Polyphenols Profile in Different Buckwheat Accessions

The identification and quantification of different polyphenols (phenolic acids and flavonoids), such as 4-chlorobenzoic acid (4-CBA), vanillic acid (VA), cinnamic acid (CA), dihydroxybenzoic acid (DA), ferulic acid (FA), p-coumaric acid (P-CA), phthalic acid (PA), syringic acid (SA), m-toluic acid (M-TA), luteolin (LU), syringaldehyde (SY), protocatechuic acid (PTA), quercetin (QE), vanillin (VN), salicylic acid (SAA), 4-hydroxyacetophenone (4-HA), catechin (CAT), epicatechin (ECAT), orientin (OR), vitexin (VIT), hypericin (HYP), and rutin (RU) showed different profiles for each of the four tested buckwheat accessions. In fact, while DA, FA, P-CA, SA, LU, QE, VN, 4-HA, ECAT, OR, VIT, and RU were present in the four buckwheat accessions (Gema, PI481671, Kora, and Eva), VA, SY, and SAA were only present in Gema, PI481671, and Eva; 4-CBA and PTA in Gema and Eva; CAT in Gema, Kora, and Eva; PA in Eva and PI481671; and HYP only in Gema, while M-TA was missed in this buckwheat accession (Table 3).

From all of the polyphenols measured, DA, P-CA, LU, 4-HA, OR, and VIT were the most commonly accumulated compounds, especially in the roots, for most of the four buckwheat accessions in the presence of both weeds, *L. rigidum* and *P. oleracea*, while DA, LU, 4-HA, CAT, ECAT, and OR were the most altered compounds in the roots, shoots, and root exudates of the different buckwheat accessions (Table 3). Although in the presence of *L. rigidum* buckwheat tended to accumulate the compounds indistinctly in roots or shoots, depending of the buckwheat accession, there was a clear trend in the four buckwheat accessions to accumulate the compounds in the roots when co-grown with *P. oleracea*, as DA, P-CA, LU, OR, and VIT were significantly accumulated in the roots in three of the four buckwheat accessions tested when *P. oleracea* was present in the medium. On the contrary, *L. rigidum* was the weed that more root exudates induced from the buckwheat accessions tested, especially for Eva (Table 3). Finally, the Tartary buckwheat accession (PI481671) showed a generally reduced root exudation of polyphenols in comparison with the three common buckwheat accessions (Gema, Kora, and Eva), which reacted more to the presence of weeds exuding more or less phenolic acids and flavonoids, but also exuding more polyphenols when growing alone.

During the comparison of the polyphenol content of shoots, roots, and root exudates of the four buckwheat varieties (Eva, Gema, Kora, and PI481671), the generated score plot of the PCA (Figure 1A) revealed a complex accession distribution pattern that, in the largest distance, was not mainly accession-dependent, but organ-dependent as the root exudates showed strong dissimilarity with shoots and roots, especially for Gema, Eva, and PI491671. The supervised PLS-DA analysis (Figure 1B) confirmed the separation, previously observed with the PCA. The separation was achieved by virtue of the first two principal components, which explained a total variance of 60.2%. Component 2 explained the highest variance (40.5%), while component 1 explained 19.7% of the total variance. The hierarchical cluster showed similarity in the root and shoot polyphenol contents of each accession, being the root and shoot contents of the four accessions grouped together in the same cluster of the dendrogram, but each species constituting and independent subclusters of the tree (Figure 1C). Surprisingly, this division was not species dependent, as there was more distance among the common accessions Eva and Gema with Kora than among the common accessions Gema and Eva with the Tartary accession PI481671.

In contrast, the root exudates of Gema, Eva, and PI481671 were grouped in a totally independent cluster, showing its dissimilarity from the rest of the samples and indicating a characteristic behaviour of exudation in buckwheat. Moreover, although the content of root exudates of common and Tartary accessions was grouped in the same cluster of the dendrogram, Gema and Eva were grouped in a different subcluster than PI481671, suggesting a species-dependent dissimilarity among accessions at this level of distance. The compounds that had stronger weight in this classification were LU, QE, 4-HA, M-TA, SAA, CAT, and PTA (over 1.0 VIP score). LU, QE, and 4-HA were characterized to be increased in Gema, Kora, and PI481671 shoots, roots, and root exudates in comparison with Eva, while M-TA was increased in Eva shoots, roots, and root exudates in comparison with Gema, Kora, and PI481671 (Figure 1D).

#### 2.2.1. Chemical Profile of EVA Polyphenols

When Eva was co-cultured with *L. rigidum*, the concentration of most of the polyphenols was significantly increased in the shoots, roots, and especially root exudates, following a co-culture of 7 days (Table 4).

Different specialized metabolites such as VA, DA, FA, *P*-CA, LU, SY, PTA, QE, VN, and 4-HA were significantly increased in the **shoot tissues** after the co-culture of Eva with *L. rigidum* (Table 4). Even VA and VN were found in high concentrations after co-growth with annual ryegrass, although could not be detected in Eva shoots when growing alone.

In contrast, there was a clear trend to reduce most of the detected polyphenols (DA, FA, *P*-CA, PA, LU, PTA, QE, SAA, CAT, ECAT, OR, VIT, and RU) in Eva shoot tissues when co-cultured with *P. oleracea*, especially LU, ECAT, and RU, which were reduced for more than 57 times, 47 times, and 31 times, respectively, followed by OR and PA, with reductions of more than 12 times. There was also a significant reduction in PTA (6.6 times), and QE (4 times). In contrast, a significant increase was only obtained in 4-CBA, VA, and 4-HA levels in the shoot tissues of Eva when co-cultured with *P. oleracea* (Table 4).

In **root tissues**, the significant increase in the content of phenolic acids and flavonoids, such as *P*-CA, LU, SY, PTA, 4-HA, CAT, ECAT, OR, and VIT when Eva was co-cultured with *L. rigidum* was coherent with the increase observed in the shoots and root exudates of this buckwheat accession in the presence of annual ryegrass. Especially interesting were the results of SY, ECAT, and VIT that increased their contents by more than 12 times, and of LU, which could not be detected in the control, although it appeared in Eva when co-growing with *L. rigidum*. In Eva roots co-cultured with *P. oleracea,* there was also an increase in VA, *P*-CA, LU, SY, PTA, CAT, ECAT, OR, and VIT following the pattern previously described for Gema and Kora. Surprisingly, LU was found to be increased after co-culture with both weeds, but was absent in the root tissues of the control plants of Eva. As a result, Eva root tissues accumulated in general significantly strong levels of polyphenols (Table 4).

In contrast, Eva **root exudates** were strongly increased only in the presence of *L. rigidum,* with 18 significantly increased exuded compounds (VA, DA, FA, *P*-CA, PA, SA, LU, SY, PTA, QE, VN, RU, SAA, 4-HA, CAT, ECAT, OR, and VIT) out of 20 compounds detected (Table 4). In fact, nine of the polyphenols significantly increased after *L. rigidum* co-culture did not practically appear in the control samples (when Eva was growing alone), which could not be detected in the analyses. Especially interesting is the exudation of PTA (2836-times), ECAT (1034-times), FA (511-times), DA (477-times), RU (467-times), CAT (346-times), VA (345-times), QE (304-times), OR (216-times), LU (200-times), VIT (81.6-times), SA (47-times), and VN (22.6-times). In contrast, in general, root exudates significantly decreased following the co-cultivation of Eva with *P. oleracea* and even the polyphenols DA, *P*-CA, and SY practically disappeared from the medium, while the content was only significantly increased for the polyphenols PTA, 4-HA, and CAT, which could not be detected in the root exudates when the Eva buckwheat plants were growing alone (Table 4).

#### 2.2.2. Chemical Profile of GEMA Polyphenols

As shown in Table 5, the distribution of polyphenols in the shoots, roots, and root exudates of Gema was different depending on whether this accession was grown with *L. rigidum* or with *P. oleracea*.

Compared with the control, where Gema plants were growing alone, only the concentration of DA was significantly increased in **shoot tissues** when Gema was co-cultured with *L. rigidum* (Table 5), while the level of PTA, SAA, and 4-HA significantly increased in the shoot tissues when Gema was co-cultured with *P. oleracea*. Although the increases in PTA and SAA were not especially relevant, 4-HA increased by 24 times its concentration in the shoot tissues in the presence of *P. oleracea*. In addition, there were more polyphenols inhibited in the shoots when Gema was co-cultured with *P. oleracea* (DA, LU, SY, VN, ECAT, OR, and VIT) than with *L. rigidum* (VN, CAT, and ECAT). Curiously, the concentration of the polyphenol VN almost disappeared in the shoot tissues when Gema was co-cultured with either weed species. Similar to the shoot analyses, Gema increased significantly more polyphenols in **root tissues** when co-cultured with *P. oleracea* (DA, *P*-CA, LU, 4-HA, RU, OR, and VIT) than with *L. rigidum*, where only 4-HA and RU showed a significantly increased content. In fact, a significantly strong accumulation in the amounts of DA (56-times), LU (98-times), OR (5-times), VIT (4-times), and RU (3-times) was found in the **roots** of Gema when co-cultured with *P. oleracea* (Table 5).

Regarding the presence of polyphenols in **root exudates**, the levels of many of the specialised metabolites decreased significantly when Gema plants were grown in co-culture with *P. oleracea* (4-CBA, DA, FA, SY, RU, CAT, and ECAT) and some of them (DA, FA, SY, RU, CAT, and ECAT) practically disappeared in Gema root exudates following co-growth with common purslane. In fact, only one polyphenol (QE) was slightly more exuded in front of common purslane than when Gema plants were grown alone. In contrast, none of the polyphenols was decreased in the presence of annual ryegrass while CAT and OR significantly increased their exudation to the medium with increases of more than five times for CAT and more than three times for OR (Table 5).

#### 2.2.3. Chemical Profile of KORA Polyphenols

When Kora was co-cultured with *L. rigidum* (Table 6), there was a general decrease in polyphenols in the roots (DA, FA, *P*-CA, LU, QE, RU, OR, and VIT) and root exudates (DA, FA, *P*-CA, SA, LU, QE, 4-HA, RU, CAT, ECAT, and VIT) that was not found for common purslane, which showed the most inhibited polyphenols in the **shoots**, with DA, FA, *P*-CA, SA, *M*-TA, LU, 4-HA, CAT, and ECAT practically disappearing from the shoot tissues, and QE, OR, RU, and VIT also significantly reduced in the presence of this weed compared with the control. As a result, no polyphenols were found to be accumulated in the shoots of Kora after co-growth with *P. oleracea*.

In contrast, once more, different polyphenols were found to be significantly accumulated in the presence of *P. oleracea* in **root tissues**, as *M*-TA (919-times), 4-HA (17-times), DA (16-times), LU (10-times), VIT (7-times), and OR (5.6-times), while most of the polyphenols, were strongly reduced after co-cultivation with *L. rigidum*, as already explained above.

This reduction in the polyphenols in root tissues of annual ryegrass was consistent with the general reduction of polyphenols found in Kora **root exudates** when co-grown with *L. rigidum*, with many of them, such as DA, FA, *P*-CA, SA, LU, 4-HA, CAT, and ECAT, practically disappearing from the root exudates following co-growth with *L. rigidum.* In contrast, Kora strongly exuded a higher amount of several phenolic acids and flavonoids in the presence of *P. oleracea*, such as *M*-TA (321-times), 4-HA (145-times), DA (10-times), FA (5.6-times), OR (5.4-times), and *P*-CA (3.2-times). Finally, OR was the only polyphenol more exuded when Kora was co-cultivated with *L. rigidum* compared with the control (Table 6).

#### 2.2.4. Chemical Profile of PI481671 Polyphenols

In general, the root and shoot tissues of PI481671 were more sensitive to the presence of *P. oleracea* than *L. rigidum* in the medium (Table 7). In fact, while only three polyphenols were altered in **PI481671 shoot tissues** in the presence of annual ryegrass (increases in *M*-TA and QE, and a decrease in PA), the content of seven polyphenols was altered in the presence of common purslane (increases in DA, FA, SAA, 4-HA, and ECAT, and decreases in VA and PA). This pattern was even more obvious in the **root tissues**, where the content of four polyphenols was altered in the presence of annual ryegrass, while the content of up to 14 polyphenols was altered in the presence of common purslane. In fact, after co-growth with *P. oleracea*, the **root tissues** of PI481671 accumulated strong amounts of polyphenols, showing a significant increase in the concentration of different specialized metabolites, such as VA, DA, FA, *P*-CA, SA, VN, SAA, 4-HA, RU, and ECAT. Especially interesting are the strong increases of ECAT (200-times), RU (26-times), and SAA (23-times). In contrast, the results only showed a significant increase in one polyphenol, OR (278-times), in the root tissues after the co-cultivation of PI481671 with *L. rigidum* (Table 7).

Although there was a general lower **root exudation** in the Tartary buckwheat accession PI481671 when compared with the common buckwheat accessions (Kora, Gema, and Eva), there was a significant increase in the root exudates of DA, SA, QE, and ECAT following the co-cultivation of PI481671 with *L. rigidum*, indicating that the levels of polyphenols in the root exudates had a propensity to rise. In contrast, only VIT increased in the root exudates after co-cultivation with *P. oleracea*.

#### 2.2.5. Multivariate Analyses of Polyphenols’ Profile of Shoots, Roots, and Root Exudates of the Four Buckwheat Accessions

When independently comparing the polyphenol profile of the shoots (Figure 2A), roots (Figure 2B) and root exudates (Figure 2C) of Gema, Eva, Kora, and PI481671, the results of the PLS-DA analysis further exacerbated the separation among accessions for both shoots and roots (Figure 2A left, Figure 2B left, respectively), but not for the root exudates. The root exudates of each accession alone or in co-culture with *L. rigidum* or *P. oleracea* were separated and, at the same time, overlapped with the samples of the other accessions (Figure 2C left). PLS-DA explained a total variance of 54.2% for shoots (component 1: 23% and component 2: 31.2%), 46.1% for roots (component 1: 29.4% and component 2: 16.7%), and 60.5% for root exudates (component 1 23.1%: and component 2: 37.4%). In fact, the polyphenol profile of the roots of the accessions Gema (G), Eva (E), Kora (K), and PI481671 (PI) were perfectly separated from each other but perfectly grouped into each accession for all of the plants growing alone (A) or co-cultured with *L. rigidum* (L) or *P. oleracea* (P) (Figure 2B left). The same was found for the shoot polyphenol profile of the accessions Gema, PI481671, and Eva (Figure 2A left). In contrast, no clear groups were found for root exudates, where the polyphenol profile of the plants of the different accessions growing alone or in co-culture with *L. rigidum* or *P. oleracea* were completely overlapped for the four accessions tested in this study (Figure 2C left).

The variable importance in projection (VIP) scores (built on the polyphenols with a VIP score higher than 1.0) revealed that PTA and *M*-TA were the only polyphenols with VIP scores higher than 1.0 that were found in all of the analyses (shoots, roots, and root exudates), while CAT and RU were also common for the shoots and roots. In particular, the compounds with a higher VIP score in the shoots were, in order of importance, CAT, PTA, RU, VN, *M*-TA, 4-HA, and OR, while in the roots they were LU, CAT, PTA, QE, *M*-TA, RU, and PA, and in the root exudates they were OR, 4-CBA, SAA, SY, PTA, PA, and *M*-TA (Figure 2A right, Figure 2B right, Figure 2C right, and Table 8).

## 3. Discussion

Crop species with allelopathic activity are known as good options for reducing weed damage in sustainable agroecosystems [39]. Although the allelochemical potential of buckwheat crops to manage weeds in the field has not been deeply studied up until now, different papers suggest the presence of bioactive compounds on their extracts and residues that can control the development of different weeds [23,26]. Moreover, different genotypes may produce specialized metabolites differently, opening a wide variety of allelopathic potentials and, consequently, suppressive effects on weeds [40,41]. For this reason, studies such as those carried out in this work, evaluating different varieties of buckwheat that, by themselves, as a crop (alive plants), can control the presence of weeds in their environment, are highly indispensable.

The current study provides further evidence that phenolic compound synthesis, distribution, and exudation vary among buckwheat accessions, and that these compounds play a role in the interference among plants of crops and weeds. Phenolic compounds are specialized metabolites that can behave as phytotoxic when exuded into the medium, affecting the growth of neighbouring plants, as reported for several phenolic acids [42]. Meanwhile, the identification of allelochemical substances and their particular mode of action and interference in different physiological processes is required to make use of the allelopathic capabilities of crop plants in weed control [43]. Additionally, previous studies revealed that production of phenolic compounds differ greatly between buckwheat accessions in the different tissues [40,44].

Phenolic compounds, and especially flavonoids, have been reported for decades as strong antioxidant compounds, behaving as protectors into the plant metabolism against any external biotic or abiotic damage to which the plant can be exposed [45,46]. In fact, flavonoids are the most reported specialized metabolites in the plant defence system [47], so they may play an important role in plant−plant competition other than allelopathy, by accumulating in the different organs of the plant (leaves, roots, stems, seeds, etc.), and make plants more resistant and resilient against external attacks [48].

Moreover, as reported by Uddin et al. [41], different buckwheat cultivars can show different contents of phenolic compounds, and even the same cultivar can show organ-related differences in the phenolic composition. Studying three common buckwheat cultivars (Suwon1, Suwon 2, and Suwon 12), they found that Suwon 1 had the highest levels of catechin and epicatechin, while the greatest amount of 4-hydroxybenzoic acid, chlorogenic acid, and 4-hydroxy-3-methoxybenzoic acid was present in the cultivar Suwon 2. In this context, the Suwon 2 cultivar dominated over the other two cultivars, with the highest phenolic compound content in the stem, flowers, and roots of common buckwheat. Something similar was found in this study, as when comparing the four accessions, the hierarchical cluster showed more dissimilarities between the samples, grouping shoots, and roots separately from the root exudates, than between species, as common and Tartary accessions were grouped together in the same branch of the dendrogram, although each accession was separated from the others in the sub-branches of the tree.

When analysing the response of the different buckwheat accessions to the weeds, our results showed that the most relevant polyphenolic compounds were DA, LU, 4-HA, CAT, ECAT, and OR, as were the compounds whose chemical profile changed more in the roots or shoots and root exudates of the different buckwheat accessions along the study.

4-HA was discovered in buckwheat root exudates and in soil extracts following buckwheat cultivation, in addition to flavonoids and phenolic acids [26]. On the other hand, CAT was found to be highly phytotoxic against *Arabidopsis thaliana* (L.) Heynh. and *Festuca idahoensis* Elmer [49]. According to previous research, the phytotoxicity of CAT on the root cell tissues of *A. thaliana* is caused by the cytoplasm condensing due to the rapid induction of reactive oxygen species, which is followed by an increase in Ca^2+^ and acidification of the cytoplasm, resulting in cell death [50]. In our results, a strong significant increase in the root exudates (346-fold higher than the control) of CAT was observed in buckwheat variety Eva when co-cultured with *L. rigidum*. Golisz et al. [25] also established the effective concentrations for lettuce to be in this range (0.4 mM). These findings suggest that CAT is highly phytotoxic, but less selective against different weeds. In a similar way, Serniak [51] showed, in a comparative allelochemical study, that ECAT exerted strong phytotoxic effects on radish seedling growth. Moreover, ECAT significantly decreased the growth of *Lepidium sativum* L. [52], and radish root growth was also inhibited in vivo as the result of the phytotoxic activity of ECAT [51].

Because of the quick evolution of resistance in *L. rigidum* and *P. oleracea*, comprehensive weed management measures including crop allelopathic varieties are required to slow down this rapid evolution and promote sustainable control [53]. Determination of the mechanism(s) associated with weed suppression is essential to determine if the use of crop varieties for allelopathic and competitive weed suppression in cereal and pseudo-cereal crops is going to provide sustainable solutions for weed management and to overcome resistance problems in weeds.

In this study, the variety Gema showed the strongest crop competitive ability against mono and dicot weeds compared with the other buckwheat accessions of Eva, Kora, and PI481671. This common buckwheat accession greatly inhibited the germination and root length of the monocot *L. rigidum* in more than 70% when compared with the control (i.e., *L. rigidum* growing alone). Strong effects of Gema were also observed on the dicot weed *P. oleracea*, where shoot and root lengths were stimulated, while no increases in fresh plant weight could be detected, resulting in longer, but much weaker, shoots and roots. Gema accumulated more DA, FA, *P*-CA, LU, 4-HA, OR, RU, and VIT in the roots when co-cultured with *P. oleracea*, while QE was the only polyphenol significantly more exuded to the medium after co-growing with this dicot weed. Previous works have reported that buckwheat varieties can accumulate polyphenols in the roots and shoots as a means of defence or protection [27,54]. Our results indicate that strong competition may be taking place between Gema and *P. oleracea*, and that *P. oleracea* might be trying to colonize more space (via longer roots) at the cost of making its roots weaker. The pressure that Gema has on *P. oleracea* can be related to the significant increase in root length of Gema in the presence of this weed, which would be competing with *P. oleracea* by colonizing the medium. In this context, flavonoid accumulation in the roots might be protecting Gema in front of this dicot weed. Root exudation represents a carbon cost to the plant [55]; therefore, the reduced root exudation of polyphenols could save energy that the buckwheat plant could use for defence or protection against *P. oleracea*. Several researchers have reported the antioxidant properties of flavonoids from different buckwheat varieties [45,56]. In this sense, the significant increase in some polyphenols in the roots and shoots of Gema plants could be protecting them from the damage induced by the presence of this dicot weed.

Allelochemical plants, such as buckwheat, have distinct mechanisms for inducing the phytotoxic effects on monocot and dicot weeds, so that biological action on the target weed differs from one weed to another [24,26]. In this sense, the behaviour of Gema with the monocot weed *L. rigidum* was totally different than with *P. oleracea*. In fact, no alterations in leaf and root weight or shoot and root length were observed in Gema plants when co-growing with *L. rigidum*. On the contrary, the germination and root length of *L. rigidum* were strongly inhibited by Gema, with 80% and 70% inhibitions, respectively. Gema increased the exudation of CAT by more than five times and OR by more than three times in the presence of *L. rigidum*, which could be enough to inhibit the germination and growth of *L. rigidum*, as there is no relevant accumulation of polyphenols on the roots or shoots of buckwheat plants and neither growth parameters of Gema plants are affected in front of this monocot weed, which suggests that *L. rigidum* does not represent a threat to Gema plants. The phytotoxic activity of CAT and OR would be enough for Gema to handle *L. rigidum* development [22].

The next accession with a strong capacity to sustainably control weeds was the Eva variety. This common buckwheat accession greatly inhibited the germination and fresh plant weight of the monocot *L. rigidum,* while it strongly stimulated the shoot length of this weed, which resulted in longer but weaker plants. Something similar was observed on the dicot *P. oleracea*, where Eva stimulated the shoot length but did not increase the plant weight. We revealed that this competitive ability of Eva was related to its robust root exudation of different polyphenols (phenolic acids and flavonoids), such as VA, DA, FA, *P*-CA, PA, SA, LU, SY, PTA, QE, VN, 4-HA, RU, CAT, ECAT, OR, and VIT, when co-cultured with both weeds; this was especially true for *L. rigidum*, which induced a significant increase in the exuded content for 16 polyphenols out of the 19 analysed. This increased the production and exudation of different polyphenols, especially PTA (2836-times), ECAT (1034-times) FA (511-times), DA (477-times), RU (467-times), CAT (346-times), VA (345-times), QE (305-times), OR (215-times), and LU (200-times) by Eva compared with the control, which would ensure the success in the inhibition of the germination and development of *L. rigidum* and in the induction of increased weakness in *P. oleracea*. Our results demonstrated that when buckwheat recognizes the presence of the weeds it subsequently changes its root exudation profile to impede their growth. These results are consistent with those found by Gfeller et al. [57] for buckwheat in the presence of redroot pigweed. Moreover, although previous studies [20,54] have suggested the accumulation and exudation of RU as the responsible allelochemical molecule to inhibit the growth of different weeds, our results showed that there are a plethora of compounds participating in this phenomenon, and that there are other polyphenols, such as PTA, ECAT, or FA, that can playing an even more strong allelochemical role than RU. In this sense, in recent research, Krumsri et al. [58] evaluated the phytotoxic potential of *Dalbergia cochinchinensis* Pierre ex Laness. and found that PTA, the most exuded compound by Eva roots, caused growth inhibition on *Echinochloa crus-galli* (L.) P. Beauv. and *L. sativum* at low concentrations. At concentrations greater than 10 mM, ECAT, the second most exuded compound by Eva, significantly decreased the growth of *L. sativum* [52]. Radish root growth was also inhibited in vivo because of the ECAT phytotoxic activity [51]. In another study, Hussain and Reigosa [43] evaluated the effects of FA and DA on the photosynthesis of *Rumex acetosa* L., and found that both compounds behaved as potent inhibitors of photosynthetic traits, leading to weaker plants. The strong increase in the root exudation by living plants indicates that Eva molecules attacked the herbicide-resistant weeds (*L. rigidum* and *P. oleracea*), inhibiting the germination of *L. rigidum* and hindering the development of both monocot and dicot weeds. The exudation of several flavonoids (QE, VN, 4-HA, CAT, ECAT, OR, RU, and VIT) demonstrates that the defence strategy of Eva is alive and working closely with the attacking phenomena to obtain access to the available resources (space and light) for its growth and development. Moreover, polyphenols can also play a role of defence and protection in the plant, as previously demonstrated by several authors [27,54]. In this context, Eva also significantly increased the production of several polyphenols in the shoots (VA, DA, FA, *P*-CA, LU, SY, PTA, QE, VN, and 4-HA) and roots (*P*-CA, LU, SY, PTA, 4-HA, CAT, ECAT, OR, and VIT) after growing with *L. rigidum*, and in root tissues (VA, *P*-CA, LU, SY, PTA, CAT, ECAT, OR, and VIT) after growing with *P. oleracea*. Especially interesting were the polyphenols *P*-CA, LU, SY, PTA, CAT, ECAT, OR, and VIT, which were found to increase in the root tissues in the presence of both weeds in very high concentrations.

Kora was the common buckwheat accession that affected the development of *L. rigidum* and *P. oleracea* less in the present study, although it showed a strong effect on the germination of both monocot and dicot weeds. Although Kora did not exude phenolic compounds and flavonoids after growth with *L. rigidum*, the accumulation in the roots and shoots of different polyphenols, such as *M*-TA, 4-HA, and OR could improve the antioxidant activity in buckwheat plants, providing an advantage in plant−plant competition [59].

In contrast, Kora exuded significant amounts of DA, FA, *P*-CA, *M*-TA, 4-HA, and OR after growth with *P. oleracea*, which were not only related to the strongest decrease in *P. oleracea* germination for the four tested accessions, but also to the chemical control of *P. oleracea* by Kora, where the weed seedlings could normally grow, without affecting the growth and development of Kora plants.

The Tartary buckwheat accession PI481671 followed a similar chemical profile and buckwheat plant development to the common buckwheat accession Gema, although this accession did not affect the weeds in a similar pattern. PI481671 stimulated the weed total biomass and shoot length of *P. oleracea* while inhibiting the total weight of *L. rigidum*. This could be explained by the results previously found by Sijahović et al. [38], who demonstrated that buckwheat−weed interactions are dependent on the type of weed present in the neighbours, resulting in changes in the exudation behaviour of buckwheat plants. The root tissues of PI481671 indeed accumulated considerable levels of polyphenols after co-growing with *P. oleracea*, displaying a marked rise in the content of many specialized metabolites, including VA, DA, FA, *P*-CA, SA, VN, SAA, RU, 4-HA, and ECAT. The significant increases in ECAT (200 times) and SAA (23 times), a well-known defence compound, are particularly fascinating. Competitive genotypes can better access light, nutrients, and water resources in limited space, thus suppressing the growth and reproduction of nearby weed species [37]. Although the Tartary buckwheat (PI481671) showed generally less root exudation than the common buckwheat accessions, PI481671 increased, as Eva, the exudation of DA, SA, and QE, by several folds after co-culture with *L. rigidum*. In a recent study, Šćepanović et al. [60] showed that strong doses of VA, DHA, and *P*-CA, as well as the phenolic acid mixture, inhibited the early growth of *Ambrosia artemisiifolia* L.

As shown by the multivariate analyses, in this study, no differences were found among species (*F. esculentum* and *F. tataricum*), but among the analysed samples. In fact, the roots and shoots were grouped separately from the root exudates. When having a look at the compounds with a higher VIP score (weight) for the comparison of the different samples, the polyphenols PTA and *M*-TA were common to all of the analyses (shoots, roots, and root exudates), while CAT and RU were also common for the shoots and roots. Most of these compounds (i.e., CAT, RU, and PTA) have been shown to have antioxidant properties [61,62,63] against different stress factors, which could be one main reason for their accumulation in the roots and shoots of common and Tartary buckwheat accessions, giving an advantage to buckwheat plants in front of the surrounding weeds.

Our findings show that different buckwheat accessions have varying capacities to release or accumulate specific metabolites in the presence of surrounding weeds, as well as varying capacities to manage those weeds sustainably. Additionally, each accession exhibited varied the inhibitory capacities and chemical profiles against monocot or dicot weeds, depending on the type of weed in their vicinity [20,25]. These results are consistent with those of Kalinova [30], who found varietal differences for the inhibition of lettuce by three different buckwheat varieties and related these differences to allelochemical action of buckwheat by measuring the production of the known allelochemical compounds ECAT and RU.

The findings of this study indicate that the buckwheat accessions that most significantly impacted the growth of the tested weeds were those with the highest production of allelopathic compounds and their exudation into the rhizosphere. The inhibitory effect on weed germination and growth could be caused by the allelochemicals that were exuded to the medium, because in the current experiments, there was no direct physical contact between the roots or shoots of the buckwheat and weeds. This was particularly true for the variety Eva, which demonstrated a high potential for controlling monocot weed *L. rigidum* trough root exudates. However, the superior competitive ability that the accumulation of polyphenols in shoots and roots provided buckwheat plants in front of weeds could be driving the negative impact of the tested buckwheat accessions on the two target weeds, even though these weeds showed resistance against different herbicides.

The present results highlight the necessity to screen different buckwheat accessions to find the better ones to be used in organic agriculture, due to the variation in the synthesis, distribution, and exudation of polyphenols, which can provide a different allelopathic or competitive ability to different accessions.

## 4. Materials and Methods

### 4.1. Germination and Growth Bioassays

Four different buckwheat accessions (Gema, Kora, Eva, and PI481671), previously pre-selected for their potential to be used in organic farming (in the frame of the EU project ECOBREED), were screened for their allelopathic potential against germination and seedling growth of common purslane (*Portulaca oleracea* L.) and annual ryegrass (*Lolium rigidum* Gaud.) using perlite as an inert substrate. Eva is a commercial variety from Kmetijski inštitut Slovenije/Agricultural Institute of Slovenia (KIS), Slovenia. Gema, Kora, and PI481671 were provided by the Czech Gene Bank—Crop Research Institute of Prague (Czech Republic).
**Identification Number****Species****Name****Origin**01Z5000095*Fagopyrum esculentum*KoraPoland01Z5000112*Fagopyrum esculentum*GemaPoland01Z5100009*Fagopyrum tataricum*PI 481671Bhutan

Seeds of common purslane and annual ryegrass were obtained commercially from “Semillas Cantueso” (Cantueso Natural Seeds, Cordoba, Spain) and Herbiseed (Herbiseed Twyford, Berkshire, UK), respectively. Surface-sterilized seeds of each buckwheat accession were grown alone in individual plastic trays (32 × 20 × 6 cm) filled with a 5 cm deep layer of perlite (500 g/tray), watered with distilled water, and kept in a growth chamber with a day and night temperature of 20 °C and 12/12 h light/dark photoperiod. The light was supplied by cool white fluorescent tubes and irradiance was maintained at 275 µmol m^−2^ s^−1^. The buckwheat seeds were sown and left to germinate for 5 days before transferring to treatment plastic trays.

Ten buckwheat seedlings per plastic tray and three plastic trays per treatment were used in these experiments for each common and Tartary buckwheat accession. At day 1, similar seedlings for each buckwheat accession were selected and sown in a plastic tray with perlite and were watered each other day. The seedlings were placed in three rows on one half of the tray and placed in a controlled environment growth cabinet with a daily photoperiod of 12L:12D and continuous temperature of 20 °C. After 10 days of growth for the buckwheat accessions, *L. rigidum* or *P. oleracea* seeds were added to the other half of the tray. The arrangement was such that the allelochemicals produced and released by the buckwheat seedlings could diffuse throughout the perlite medium to influence weed germination and growth, but no physical contact was allowed among the roots or shoots of buckwheat seedlings and weeds.

After growing together (buckwheat and weeds) under the same conditions for 7 days, the germination rate, total weight, shoot and root length, and plant height were measured in the two target weeds, *L. rigidum* and *P. oleracea*.

The five treatments established for each buckwheat accession were as follows:(1)Buckwheat alone (Gema, Kora, Eva, or PI481671);(2)Buckwheat seedlings (10 days germinated) + *L. rigidum* (10 ungerminated seeds);(3)Buckwheat seedlings (10 days germinated) + *P. oleracea* (10 ungerminated seeds);(4)*P. oleracea* alone;(5)*L. rigidum* alone.

### 4.2. Plant Harvest and Metabolite Extraction

After harvesting the shoots and roots of the buckwheat seedlings and collecting the root exudates from each treatment tray, the samples were stored at −80 °C until extraction. The samples from each buckwheat accession were separately processed for the identification and quantification of phenolic compounds and flavonoids from the shoots, roots, and root exudates. The plant tissues (shoots and roots) were lyophilized, ground into powder with a mortar and pestle after the addition of liquid nitrogen, and macerated with 9 mL of HCl (1 mM). Afterwards, the whole solution was transferred into a vial, sonicated for 15 min (Branson SINIFIER 250; microTip limit, output 3), and centrifuged at 20,000 rpm at 10 °C for 15 min (SORVALL RC 5B Plus, Du Pont). The supernatant was collected and extracted three times with diethyl ether (DE). The aqueous layers were discarded, and the corresponding organic layers were combined. The organic phases were evaporated in a multivapor (P-12; Buchi, Switzerland) with 12 simultaneous evaporating positions. The multivapor (P-12) comprised of a vacuum pump (V-700), vacuum controller (V-850), rotavapor (R-210), heating bath (B-491), and recirculating chiller (F-105). The temperature of the recirculating chiller was set to −10 °C. The organic layers (DE) were placed in 15 mL plastic tubes and attached to the multivapor to evaporate the organic solvent under reduced pressure (456 mbar for DE) at 35 °C. Because of the chilling temperature (−10 °C) of recirculating chiller, and the temperature of the heating bath (35 °C), the organic solvent was evaporated and condensed in the attached crystal balloon. The final volume of the residual solution was approximately 1 mL and this solution was further dried with N_2_. Methanol was used to dissolve the residual powder and was injected for the LC-MS analysis.

The perlite-based nutrient-free water-growth medium was collected and adjusted to pH 3.0 with 0.06 M HCl. Then, 25 mL of the root exudated water was extracted three times with 25 mL DE. Further treatment of the root exudated water samples was identical to the preparation of shoots or roots extraction as described above.

### 4.3. Identification of Phenolic Compounds by LC-MS

Shoots, roots, and root exudates were extracted from each buckwheat accession using diethyl ether. Specialized metabolites (phenolic acids and flavonoids) were separated on ultra-high performance liquid chromatography coupled to a quadrupole time-of-flight high-definition mass spectrometry detector (UHPLC-qTOF-MS, Thermo Fisher Scientific Inc., Madrid, Spain) according to Wu et al. [64] with small slight amendments according to Hussain et al. [65].

High Performance Liquid Chromatography−Mass Spectrometry (HPLC-MS, 1260 Series, Agilent, Santa Clara, CA, USA) was performed using a system consisted of compact mass detector equipment (TRIPLE QUAD 3500; AB SCIEX INSTRUMENTS, AB Sciex Pte. Ltd., Framingham, MA, USA). Polyphenols were separated with a C18 column (PHENOMENEX LUNA, 150 mm × 2 mm and 3 μm, Phenomenex, Inc., Torrance, CA, USA) using different chromatographic conditions depending of the compounds. Hypericin was separated at a flow rate of 400 μL min^−1^ with a column temperature of 40 °C, an injection volume of 10 μL, and the column was equilibrated for 6 min between runs. The isocratic elution used was a mixture of two solvents: A, consisting of 5 mM ammonium acetate and 0.1% acetic acid in water, and B, consisting of acetonitrile. The isocratic conditions were 25% A and 75% B for 10 min. The other phenolic acids were separated at a flow rate of 300 μL min^−^^1^, the column temperature was 40 °C, the injection volume was 10 μL, and the column was equilibrated for 6 min between runs. The gradient elution used was a mixture of two solvents: A, consisting of 0.1% formic acid in water, and B, consisting of 0.1% formic acid in acetonitrile. Initial conditions (98% A and 2% B) were held for 4 min before ramping to 20% B at 7 min and 90% B at 14 min. Initial conditions were recovered at 15 min and held until 21 min. The instrument parameters were as follows: curtain gas (CUR), 25 psi; collision gas (CAD), 7 psi: ion spray voltage (IS), −4500 V; temperature (TEM), 400 °C; ion source gas 1 (GS1), 55 psi; ion source gas 2 (GS2), 55 psi; interface heater, on.

The quantification of the concentration of the compounds was obtained from calibration curves that related the detector’s response to the pure analyte’s concentration of those compounds identified in the chemical analyses.

### 4.4. Data Analyses

The experiments were carried out using a completely randomized design with three replications (each replication was a bulk of 10 buckwheat plants and 10 weed plants). IBM SPSS software (SPSS Inc., Chicago, IL, USA, version 22.0) was used to analyse the data. To detect outliers, an exploratory data analysis was performed. The Kolmogorov−Smirnov test was used to check for deviation from normality, and the Levene test was used to check for homogeneity. Depending on the homoscedasticity of the samples, one-way ANOVA or Kruskal−Wallis tests were performed for germination and seedling growth data to establish the significant effect (*p* ≤ 0.05) of the treatments (different accessions). The results are presented in the tables as the percentage of increase or decrease when compared with the control. Different letters represent significant differences in treatment. Polyphenol (phenolic compounds and flavonoids) data were analysed through analysis of variance and the Duncan multiple range test was performed to establish the significant effect (*p* ≤ 0.05) within the treatments (alone, co-cultured with *L. rigidum*, or co-cultured with *P. oleracea*).

The identified polyphenols were analysed using the Metaboanalyst 5.0 software. The missing values were replaced with half of the minimum value found, and then data were Log_10_ transformed and Pareto scaled. Data were then analysed through unsupervised principal component analysis (PCA), to visualize group discrimination, and through the supervised partial least square discriminant analysis (PLS-DA). Feature selection with the highest discriminatory power was based on their variable importance in projection (VIP) score *>* 1.0. To avoid overfitting, the PLS-DA model was validated using Q2 as a performance measure, the 10-fold cross-validation and setting in the permutation test a permutation number of 20 (see tables reported in Appendix A).

## 5. Conclusions

We conclude that the selection, evaluation, and development of buckwheat accessions with an increased competitive ability and strong allelopathic potential will be a good option for sustainable weed management. Our results confirm the ability of different buckwheat accessions to suppress monocot and dicot weeds, and this ability clearly appears to be accession dependent. In this regard, Gema appears to be the accession that should be used for growing in organic agriculture due to its capacity to sustainably regulate the germination and growth of the monocot weed *L. rigidum* and the dicot weed *P. oleracea,* while stimulating the root growth of buckwheat plants. Meanwhile, all four buckwheat accessions showed varying degrees of allelochemical production and release to control weeds through affecting multiple processes, such as germination, growth, and weed biomass. We conclude that different buckwheat genotypes may have different capacities to produce and exude several types of specialized metabolites, which lead to a wide range of allelopathic and defence functions in the agroecosystem to sustainably manage the growing weeds in their vicinity.

## Figures and Tables

**Figure 1 plants-12-02401-f001:**
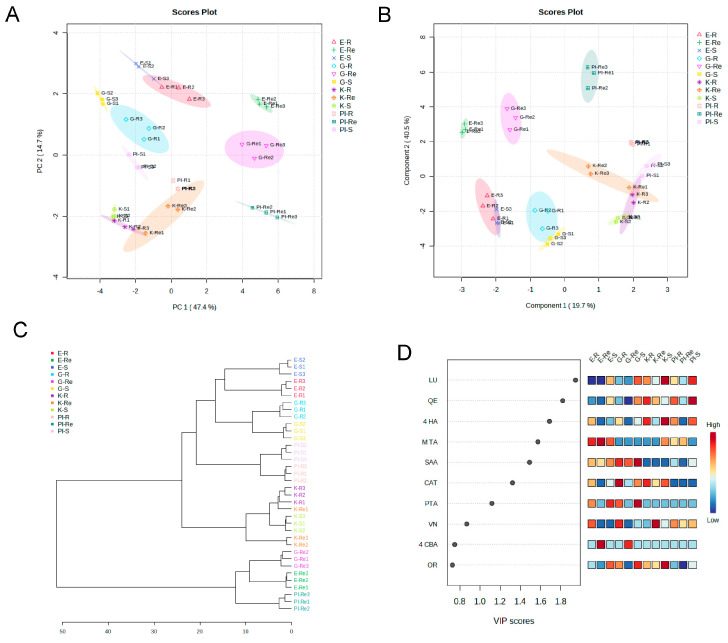
Analyses of the polyphenol content in the shoots (S), roots (R), and root exudates (Re) in the three common buckwheat accessions Eva (E), Gema (G), and Kora (K) and the Tartary buckwheat accession PI481671 (PI). (**A**) PCA scores plot between the selected PCs; (**B**) PLS-DA scores plot between the selected PCs, the explained variances of PCA and PLS-DA are shown in brackets; (**C**) clustering result shown as a dendrogram (distance measured using Euclidean, and clustering algorithm using ward). (**D**) Important features identified by PLS-DA. E−S (chemical profile of Eva shoots; dark blue), E-R (chemical profile of Eva roots; red), E-Re (chemical profile of Eva root exudates; dark green), G-S (chemical profile of Gema shoots; yellow), G-R (chemical profile of Gema roots; light blue), G-Re (chemical profile of Gema root exudates; dark pink), K-S (chemical profile of Kora shoots; light green), K-R (chemical profile of Kora roots; violet), K-Re (chemical profile of Kora root exudates; orange), PI-S (chemical profile of PI481671 shoots; malva), PI-R (chemical profile of PI481671 roots; light pink), and PI-Re (chemical profile of PI481671 root exudates; dark green). The coloured boxes on the right indicate the relative concentrations of the corresponding polyphenol in each group under study. N = 3.

**Figure 2 plants-12-02401-f002:**
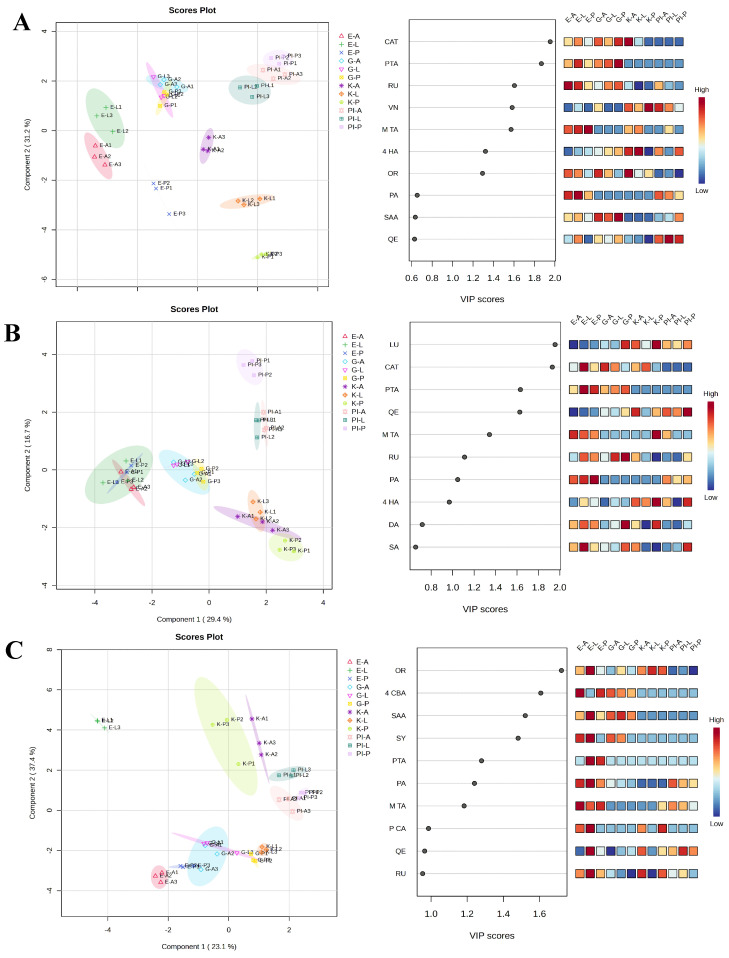
Partial least square discriminant analysis—(PLS-DA) multivariate analysis (left, scores plot between the selected PCs, the explained variances of PLS-DA are shown in brackets) and variable importance of projection (VIP) features (right, the coloured boxes on the right indicate the relative concentrations of the corresponding metabolite in each group under study) for the groups from the PLS-DA analysis, of the polyphenols’ profile in the (**A**) shoots, (**B**) roots, and (**C**) root exudates of the three common buckwheat accessions Eva (E), Gema (G), and Kora (K) and the Tartary accession PI481671 (PI) growing alone (A) or in co-culture with *L. rigidum* (L) or *P. oleracea* (P). E-A (Eva growing alone; red colour), G-A (Gema growing alone; light blue), K-A (Kora growing alone; violet), PI-A (PI481671 growing alone; light pink), E-L (Eva growing with *Lolium*; dark green), G-L (Gema growing with *Lolium*; dark pink), K-L (Kora growing with *Lolium*; orange), PI-L (PI481671 growing with *Lolium*; blue green), E-P (Eva growing with *Portulaca*; dark blue), G-P (Gema growing with *Portulaca*; yellow), K-P (Kora growing with *Portulaca*; light green), and PI-P (PI481671 growing with *Portulaca*; malva). N = 3.

**Table 1 plants-12-02401-t001:** Allelopathic impact of different buckwheat genotypes (Gema, Kora, Eva, and PI481671) on the germination (Germ: number of seeds as % of the control), fresh plant weight (PW; g given as % of the control) (i.e., weed species grown alone), and shoot and root lengths (cm given as % of the control) in monocot (*Lolium rigidum*) and dicot (*Portulaca oleracea*) weeds. Grey cells indicate significant inhibition, while bold numbers indicate significant stimulations compared with the control. The asterisk indicates statistical significant differences compared to the control at *p* < 0.05.

	*Lolium rigidum*				*Portulaca oleracea*			
*Accession*	Germ	PW	SL	RL	Germ	PW	SL	RL
PI481761	35 *	68 *	106	78	55 *	**162 ***	**186 ***	134
Eva	50 *	52 *	**149 ***	122	73	107	**146 ***	90
Kora	54 *	73 *	127	119	41 *	91	100	126
Gema	19 *	79	135	29 *	59 *	107	**158 ***	**166 ***

**Table 2 plants-12-02401-t002:** Mean values and standard deviation of growth parameters for different buckwheat accessions (Gema, Kora, Eva, and PI481671) when growing alone or in co-culture with the monocot weed *Lolium rigidum* (LR) or the dicot weed *Portulaca oleracea* (PO). The parameters measured are plant and root weight (g), and shoot and root lengths (cm). Grey cells indicate significant inhibition, while bold numbers indicate significant stimulations compared with the control.

		Leaf Weight	Root Weight	Shoot Length	Root Length
Eva	Alone	3.12 ± 0.70	1.01 ± 0.24	13.3 ± 2.09	27.2 ± 0.50
	LR	2.97 ± 0.54	0.71 ± 0.10	12.8 ± 3.11	20.6 ± 4.95
	PO	2.43 ± 0.21	0.87 ± 0.02	9.84 ± 0.83	17.3 ± 3.19
Gema	Alone	1.68 ± 0.22	0.37 ± 0.15	9.81 ± 1.83	13.8 ± 1.82
	LR	1.65 ± 0.17	0.53 ± 0.10	9.14 ± 3.97	14.9 ± 2.73
	PO	1.76 ± 0.34	**0.59** ± **0.05**	11.9 ± 2.9	**20.3** ± **2.85**
Kora	Alone	2.69 ± 0.25	1.20 ± 0.28	12.6 ± 1.07	21.4 ± 0.62
	LR	1.99 ± 1.04	0.50 ± 0.20	10.7 ± 4.33	18.7 ± 6.58
	PO	1.91 ± 0.44	0.76 ± 0.19	12.5 ± 2.46	19.6 ± 1.27
PI481761	Alone	0.25 ± 0.07	0.56 ± 0.34	4.18 ± 1.12	6.63 ± 2.36
	LR	0.33 ± 0.10	0.43 ± 0.29	4.04 ± 0.04	8.07 ± 2.26
	PO	**0.95** ± **0.07**	0.46 ± 0.17	4.43 ± 0.60	**12.7** ± **4.56**

**Table 3 plants-12-02401-t003:** Significant increases and decreases (*p* ≤ 0.05) for different polyphenols (PP) in the shoots, roots, or root exudates of the buckwheat accessions Gema (G), Kora (K), Eva (E), and PI481671 (PI) after co-culture with *L. rigidum* (LR) or *P. oleracea* (PO) compared to the control (each accession growing alone). The PP detected were 4-chlorobenzoic acid (4-CBA), vanillic acid (VA), dihydroxybenzoic acid (DA), ferulic acid (FA), *p*-coumaric acid (*P*-CA), syringic acid (SA), luteolin (LU), syringaldehyde (SY), protocatechuic acid (PTA), quercetin (QE), rutin (RU), vanillin (VN), salicylic acid (SAA), 4-hydroxyacetophenone (4-HA), catechin (CAT), epicatechin (ECAT), orientin (OR), vitexin (VIT), phthalic acid (PA), *m*-toluic acid (*M*-TA), and hypericin (HYP).

INCREASED		4-CBA	VA	DA	FA	*P*-CA	SA	LU	SY	PTA	QE	RU
Co-cultured	Shoots		E	G; E	E	E		E	E	E	E; PI	
with LR	Roots					E		E	E	E		G
	Root exudates		E	E; PI	E	E	E; PI	E	E	E	E; PI	E
Co-cultured	Shoots	E	E	PI	PI					G		
with PO	Roots		E; PI	G; K; PI	G; PI	G; E; PI	PI	G; K; E	E; PI	E	G	G; PI
	Root exudates			K	K	K				E	G	
INCREASED		VN	SAA	4-HA	CAT	ECAT	OR	VIT	PA	*M*-TA	HYP	
Co-cultured	Shoots	E		K; E						K; PI		
with LR	Roots			G; E	E	E	E; PI	E				
	Root exudates	E	E	E	G; E	G; E; PI	G; K; E	E	E			
Co-cultured	Shoots		G; PI	G; E; PI		PI						
with PO	Roots	PI	PI	G; K; PI	E	E; PI	G; K; E	G; K; E		K		
	Root exudates			K; E	E		K	PI		K		
DECREASED		4-CBA	VA	DA	FA	*P*-CA	SA	LU	SY	PTA	QE	RU
Co-cultured	Shoots			K		K	K				K	K
with LR	Roots			K	K	K		K			K	K
	Root exudates	E		K	K	K	K	K			K	K
Co-cultured	Shoots		PI	G; K; E	K; E	K; E	K	G; K; E	G	E	K; E	E; K
with PO	Roots						K					K
	Root exudates	G;		G; E	G	ECAT	PI		G; E		K	G
DECREASED		VN	SAA	4-HA	CAT	ECAT	OR	VIT	PA	*M*-TA	HYP	
Co-cultured	Shoots	G		PI	G; K;	G; K; E						
with LR	Roots			PI			G; K	G; K	PI	PI		
	Root exudates	PI		K	K	K		K	PI	E		
Co-cultured	Shoots	G	E	K	K; E	G; K; E	G; K; E	G; K; E	E; PI	K		
with PO	Roots	G			G; K	G; K	PI		PI	PI		
	Root exudates	K; PI			K; G	G			PI			

**Table 4 plants-12-02401-t004:** Quantification of polyphenols (PP) from shoots, roots, and root exudates from buckwheat accession Eva, alone (control) or in association with *L. rigidum* (LR) or *P. oleracea* (PO). Quantities are expressed in μg kg^−1^ dry weight (shoots and roots) and μg L^−1^ (root exudates). Means followed by different lowercase letters within a column show significant differences among treatments (*p* ≤ 0.05). Bold numbers indicate a significant increase when compared with the control, while grey cells indicate a significant decrease.

Sample	Treatment	4-CBA	VA	DA	FA	*P*-CA	PA	SA
Shoot	Alone	<0.01 b	<0.01 b	2472 b	83.41 b	553.2 b	350.9 a	3.187 a
	+*LR*	<0.01 b	**238.8 a**	**3956 a**	**289.1 a**	**998.4 a**	539.4 a	28.13 a
	+*PO*	**5.173 a**	**108.5 a**	131.4 c	28.69 c	98.88 c	31.55 b	6.337 a
Root	Alone	<0.01 a	471.9 b	928.9 a	152.8 a	49.71 b	147.4 a	291.0 a
	+*LR*	<0.01 a	892.9 ab	1122 a	242.9 a	**122.6 a**	186.2 a	578.7 a
	+*PO*	<0.01 a	**1067 a**	1214 a	326.9 a	**145.0 a**	218.1 a	222.5 a
Root exudates	Alone	4.533 a	<0.01 b	14.47 b	<0.01 b	123.4 b	28.02 b	<0.01 b
	+*LR*	<0.01 b	**345.5 a**	**6924 a**	**511.3 a**	**1171 a**	**448.0 a**	**47.23 a**
	+*PO*	3.980 a	<0.01 b	<0.01 c	<0.01 b	<0.01 c	10.83 b	<0.01 b
Sample	Treatment	*M*-TA	LU	SY	PTA	QE	VN	RU
Shoot	Alone	418.8 a	57.66 b	3.513 b	366.8 b	1174 b	<0.01 b	9011 a
	+*LR*	553.5 a	**205.5 a**	**14.07 a**	**1384 a**	**3211 a**	**1.948 a**	9164 a
	+*PO*	775.9 a	1.031 c	2.987 b	55.64 c	299.4 c	<0.01 b	283.2 b
Root	Alone	855.3 a	<0.01 b	2.658 b	219.1 b	122.0 a	106.1 a	409.9 a
	+*LR*	775.1 a	**3.070 a**	**31.83 a**	**2105 a**	160.6 a	197.2 a	899.9 a
	+*PO*	627.2 a	**1.810 a**	**33.92 a**	**1589 a**	173.7 a	116.2 a	791.9 a
Root exudates	Alone	1089 a	<0.01 b	2.143 b	<0.01 c	12.54 b	<0.01 b	29.08 b
	+*LR*	323.0 b	**199.9 a**	**10.54 a**	**2837 a**	**3817 a**	**22.61 a**	**13583 a**
	+*PO*	900.2 a	<0.01 b	<0.01 c	**3.710 b**	30.44 b	<0.01 b	23.66 b
Sample	Treatment	SAA	4-HA	CAT	ECAT	OR	VIT	
Shoot	Alone	25.94 a	0.603 b	260.5 a	3143 a	4220 a	3078 a	
	+*LR*	42.41 a	**2.590 a**	365.4 a	1573 b	3547 a	3504 a	
	+*PO*	16.16 b	**2.958 a**	41.35 b	67.32 c	314.7 b	549.9 b	
Root	Alone	23.76 a	26.59 b	532.3 b	33.95 b	34.92 b	30.78 b	
	+*LR*	35.64 a	**137.3 a**	**2616 a**	**414.5 a**	**241.9 a**	**449.0 a**	
	+*PO*	25.20 a	59.02 b	**1782 a**	**291.6 a**	**299.3 a**	**430.0 a**	
Root exudates	Alone	18.20 b	<0.01 b	<0.01 c	<0.01 b	20.36 b	54.87 b	
	+*LR*	**39.66 a**	**5.327 a**	**346.1 a**	**1034 a**	**4400 a**	**4481 a**	
	+*PO*	17.34 b	**6.415 a**	**2.110 b**	<0.01 b	14.40 b	24.27 b	

4-chlorobenzoic acid (4-CBA); vanillic acid (VA); dihydroxybenzoic acid (DA); ferulic acid (FA); *p*-coumaric acid (*P*-CA); syringic acid (SA); *m*-toluic acid (*M*-TA); luteolin (LU); syringaldehyde (SY); protocatechuic acid (PTA); quercetin (QE); vanillin (VN); rutin (RU); salicylic acid (SAA); 4-hydroxyacetophenone (4-HA); catechin (CAT); epicatechin (ECAT); orientin (OR); vitexin (VIT).

**Table 5 plants-12-02401-t005:** Quantification of polyphenols (PP) from shoots, roots, and root exudates from buckwheat accession Gema, alone (control) or in association with *L. rigidum* (LR) or *P. oleracea* (PO). Quantities are expressed in μg kg^−1^ dry weight (shoots and roots) and μg L^−1^ (root exudates). Means followed by different lowercase letters within a column show significant differences among treatments (*p* ≤ 0.05). Bold numbers indicate a significant increase when compared with the control, while grey cells indicate a significant decrease.

Sample	Treatm.	4-CBA	VA	DA	FA	*P*-CA	SA	PA
Shoot	Alone	<0.01 a	432.7 a	22123 b	1004 a	1317 a	92.40 a	<0.01 a
	+*LR*	<0.01 a	499.3 a	**33720 a**	933.7 a	944.7 a	131.7 a	<0.01 a
	+*PO*	<0.01 a	395.1 a	417.3 c	605.0 a	2057 a	132.7 a	<0.01 a
Root	Alone	<0.01 a	232.5 ab	286.2 b	125.3 b	35.47 b	223.3 a	<0.01 a
	+*LR*	<0.01 a	343.1 a	352.8 b	203.7 ab	53.13 b	197.0 a	<0.01 a
	+*PO*	<0.01 a	140.4 b	**16210 a**	**297.7 a**	**103.0 a**	276.3 a	<0.01 a
Root exudates	Alone	2.713 ab	<0.01 a	3.397 a	7.573 a	<0.01 a	<0.01 a	2.710 ab
	+*LR*	1.133 a	<0.01 a	2.770 a	3.797 ab	<0.01 a	<0.01 a	1.130 a
	+*PO*	0.640 b	<0.01 a	<0.01 b	<0.01 b	<0.01 a	<0.01 a	0.640 b
Sample	Treatm.	LU	SY	PTA	QE	VN	SAA	RU
Shoot	Alone	146.8 a	19.64 a	593.7 b	2208 ab	22.73 a	183.7 b	399.8 a
	+*LR*	94.40 a	17.77 a	636.3 b	1447 b	<0.01 b	113.8 b	544.9 a
	+*PO*	2.870 b	<0.01 b	**2827 a**	2720 a	<0.01 b	**265.0 a**	411.1 a
Root	Alone	2.517 b	21.00 a	244.0 a	293.0 ab	159.9 a	37.53 a	475.0 b
	+*LR*	2.193 b	34.80 a	236.3 a	247.7 b	47.73ab	43.73 a	**1275 a**
	+*PO*	**245.7 a**	38.23 a	490.0 a	454.3 a	33.77 b	48.40 a	**1617 a**
Root exudates	Alone	0.560 ab	2.331 a	<0.01 a	12.73 b	<0.01 a	26.47 a	16.41 a
	+*LR*	0.420 b	1.227 a	<0.01 a	13.77 b	<0.01 a	29.47 a	17.64 a
	+*PO*	1.070 a	<0.01 b	<0.01 a	**28.77 a**	<0.01 a	25.33 a	<0.01 b
Sample	Treatm.	4-HA	CAT	ECAT	OR	VIT	HYP	
Shoot	Alone	9.240 b	1070 a	951.8 a	4642 a	7825 a	0.097 a	
	+*LR*	14.15 b	268.5 b	61.66 c	3019 a	6542 a	0.120 a	
	+*PO*	**224.0 a**	1842 a	302.0 b	259.1 b	779.9 b	0.093 a	
Root	Alone	14.14 c	2442 a	386.6 a	697.1 b	1619 b	0.033 a	
	+*LR*	**44.00 a**	2134 a	429.7 a	97.16 c	279.1 c	0.050 a	
	+*PO*	**25.96 b**	209.2 b	45.44 b	**3610 a**	**6859 a**	0.073 a	
Root exudates	Alone	<0.01 a	3.910 b	3.197 a	5.523 b	19.05 a	<0.01 a	
	+*LR*	0.167 a	**21.52 a**	4.213 a	**16.51 a**	43.40 a	<0.01 a	
	+*PO*	<0.01 a	<0.01 c	<0.01 b	6.490 b	21.14 a	<0.01 a	

4-chlorobenzoic acid (4-CBA); vanillic acid (VA); dihydroxybenzoic acid (DA); ferulic acid (FA); *p*-coumaric acid (*P*-CA); syringic acid (SA); luteolin (LU); syringaldehyde (SY); protocatechuic acid (PTA); quercetin (QE); vanillin (VN); salicylic acid (SAA); rutin (RU); 4-hydroxyacetophenone (4-HA); catechin (CAT); epicatechin (ECAT); orientin (OR); vitexin (VIT); hypericin (HYP).

**Table 6 plants-12-02401-t006:** Quantification of polyphenols (PP) from shoots, roots, and root exudates from buckwheat accession Kora, alone (control) or in association with *L. rigidum* (LR) or *P. oleracea* (PO). Quantities are expressed in μg kg^−1^ dry weight (shoots and roots) and μg L^−1^ (root exudates). Means followed by different lowercase letters within a column show significant differences among treatments (*p* ≤ 0.05). Bold numbers indicate a significant increase when compared with the control, while grey cells indicate a significant decrease.

Sample	Treatment	DA	FA	*P*-CA	SA	*M*-TA	LU	QE
Shoot	Alone	1967 a	173.2 a	830.9 a	180.6 a	39.94 b	206.9 a	1022 a
	+*LR*	490.0 b	168.5 a	290.0 b	<0.01 b	**365.5 a**	458.6 a	406.8 b
	+*PO*	<0.01 c	<0.01 b	<0.01 c	<0.01 b	<0.01 c	<0.01 b	14.97 c
Root	Alone	476.3 b	245.3 a	659.2 a	342.1 a	<0.01 b	111.1 b	4785 a
	+*LR*	<0.01 c	21.86 b	35.86 b	147.1 a	<0.01 b	22.04 c	476,5 b
	+*PO*	**7609 a**	230.6 a	674.9 a	<0.01 b	**919.2 a**	**1178 a**	3124 ab
Root exudates	Alone	239.8 b	42.34 b	90.87 b	41.42 a	<0.01 b	18.23 a	2296 a
	+*LR*	<0.01 c	<0.01 c	<0.01 c	<0.01 b	<0.01 b	<0.01 b	13.77 c
	+*PO*	**2373 a**	**238.0 a**	**289.9 a**	24.96 a	**321.1 a**	13.32 a	109.5 b
Sample	Treatment	VN	4-HA	CAT	ECAT	OR	VIT	RU
Shoot	Alone	42.96 a	13,541 b	2230 a	516.8 a	4788 a	5632 a	176.5 a
	+*LR*	34.38 a	**33,077 a**	18.25 b	<0.01 b	2589 ab	4121 a	<0.01 b
	+*PO*	57.69 a	<0.01 c	<0.01 c	<0.01 b	2436 b	31.52 b	1.791 b
Root	Alone	23.95 ab	2214 b	2440 a	463.8 a	472.1 b	793.9 b	1041 a
	+*LR*	65.56 a	2199 b	2264 a	513.6 a	36.16 c	39.10 c	64.32 b
	+*PO*	25.70 b	**38,173 a**	64.30 b	8.760 b	**2650 a**	**5490 a**	<0.01 b
Root exudates	Alone	134.5 a	1412 b	462.6 a	68.51 a	283.0 b	224.2 a	109.1 a
	+*LR*	106.7 ab	<0.01 c	<0.01 c	<0.01 b	**3362 a**	19.57 b	<0.01 b
	+*PO*	66.41 b	**205,000 a**	229.9 b	32.26 a	**1531 a**	269.7 a	132.3 a

Dihydroxybenzoic acid (DA); ferulic acid (FA); *p*-coumaric acid (*P*-CA); syringic acid (SA); *m*-toluic acid (*M*-TA); luteolin (LU); quercetin (QE); vanillin (VN); 4-hydroxyacetophenone (4-HA); catechin (CAT); epicatechin (ECAT); orientin (OR); vitexin (VIT); rutin (RU).

**Table 7 plants-12-02401-t007:** Quantification of polyphenols (PP) from the shoots, roots, and root exudates from buckwheat accession PI481671, alone (control) or in association with *L. rigidum* (LR) or *P. oleracea* (PO). Quantities are expressed in μg kg^−1^ dry weight (shoots and roots) and μg L^−1^ (root exudates). Means followed by different lowercase letters within a column show significant differences among treatments (*p* ≤ 0.05). Bold numbers indicate a significant increase when compared with the control while grey cells indicate a significant decrease.

Sample	Treatment	VA	DA	FA	*P*-CA	PA	SA
Shoot	Alone	911.4 a	1703 b	190.4 b	355.5 ab	74.54 a	112.2 a
	+*LR*	930.7 a	1597 b	190.1 b	293.6 b	61.38 a	136.6 a
	+*PO*	540.5 b	**5940 a**	**480.0 a**	472.6 a	8.597 b	79.17 a
Root	Alone	325.9 b	97.73 b	27.52 b	17.08 b	20.91 a	78.81 b
	+*LR*	368.4 b	71.74 b	71.02 b	22.65 b	3.853 b	84.35 b
	+*PO*	**2146 a**	**276.8 a**	**283.2 a**	**148.9 a**	5.613 b	**360.9 a**
Root exudates	Alone	<0.01 a	<0.01 b	<0.01 a	<0.01 a	26.71 a	<0.01 b
	+*LR*	<0.01 a	**18.48 a**	<0.01 a	<0.01 a	3.300 b	**51.82 a**
	+*PO*	<0.01 a	<0.01 b	<0.01 a	<0.01 a	2.660 b	<0.01 b
Sample	**Treatment**	** *M* ** **-TA**	**LU**	**SY**	**QE**	**VN**	**SAA**
Shoot	Alone	<0.01 b	144.9 a	19.62 a	16,993 b	49.77 a	9.467 b
	+*LR*	**3.497 a**	115.5 a	11.41 a	**19,610 a**	37.03 a	14.95 b
	+*PO*	<0.01 b	85.45 a	12.37 a	19,429 ab	25.56 a	**69.19 a**
Root	Alone	13.28 a	21.93 a	7.620 b	4372 a	72.55 b	5.537 b
	+*LR*	<0.01 b	23.68 a	7.293 b	3850 a	37.30 b	8.903 b
	+*PO*	<0.01 b	32.50 a	**30.95 a**	5581 a	**417.6 a**	**131.5 a**
Root exudates	Alone	33.09 a	4.397 a	<0.01 a	535.7 b	57.48 a	<0.01 a
	+*LR*	18.50 a	3.793 a	<0.01 a	**1668 a**	<0.01 b	<0.01 a
	+*PO*	13.27 a	2.467 a	<0.01 a	546.5 b	<0.01 b	<0.01 a
Sample	Treatment	4-HA	ECAT	OR	VIT	RU	
Shoot	Alone	1756 b	330.0 b	98.37 ab	1786 ab	905.3 a	
	+*LR*	1.177 c	175.0 b	101.9 a	2236 a	105.2 a	
	+*PO*	**4260 a**	**6603 a**	38.71 b	1311 b	39.05 a	
Root	Alone	1754 b	23.50 b	22.12 b	193.5 a	57.65 b	
	+*LR*	0.907 c	19.83 b	**6157 a**	229.5 a	31.10 b	
	+*PO*	**4461 a**	**4807 a**	7.701 c	167.1 a	**327.5 a**	
Root exudates	Alone	<0.01 a	1.543 b	0.274 ab	22.57 b	15.84 a	
	+*LR*	<0.01 a	**401.3 a**	0.586 a	25.21 b	18.83 a	
	+*PO*	<0.01 a	1.247 b	<0.01 b	**4404** a	6.040 a	

Vanillic acid (VA); dihydroxybenzoic acid (DA); ferulic acid (FA); *p*-coumaric acid (*P*-CA); phthalic acid (PA); syringic acid (SA); *m*-toluic acid (*M*-TA); luteolin (LU); syringaldehyde (SY); quercetin (QE); vanillin (VN); salicylic acid (SAA); 4-hydroxyacetophenone (4-HA); epicatechin (ECAT); orientin (OR); vitexin (VIT); rutin (RU).

**Table 8 plants-12-02401-t008:** Variable importance of projection (VIP) features for the groups from the PLS-DA analysis for roots, shoots, and root exudates of the three common buckwheat accessions Eva (E), Gema (G), and Kora (K), and the Tartary accession PI481671 (PI) growing alone or in co-culture with *L. rigidum* or *P. oleracea*. The compounds included in the table are those compounds with a VIP score higher than 1.0 for each type of sample (shoots, roots, and root exudates). Bold numbers indicate compounds with important VIP score in the shoots, roots, and root exudates, while italic numbers indicate compounds with significant VIP score in the shoots and roots.

	Shoots	Roots	Root Exudates
CAT	*1.954*	*1.929*	
PTA	** *1.868* **	** *1.632* **	**1.279**
RU	*1.605*	*1.111*	
VN	1.582		
*M*-TA	** *1.572* **	** *1.343* **	**1.183**
4-HA	1.321		
OR	1.293		1.722
LU		1.955	
QE		1.626	
PA		1.047	1.24
4-CBA			1.607
SAA			1.521
SY			1.481

Catechin (CAT); protocatechuic acid (PTA); rutin (RU); vanillin (VN); *m*-toluic acid (*M*-TA); 4-hydroxyacetophenone (4-HA); orientin (OR); luteolin (LU); quercetin (QE); phthalic acid (PA); 4-chlorobenzoic acid (4-CBA); salicylic acid (SAA); syringaldehyde (SY).

## Data Availability

The raw data have been deposited at Zenodo webpage (https://zenodo.org/communities/ecobreed/ (accessed on 16 May 2023)), with the following DOI: https://doi.org/10.5281/zenodo.8053668 (accessed on 16 May 2023), as per the rules and regulation.

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
