# Peer review of "Specialized Metabolites Accumulation Pattern in Buckwheat Is Strongly Influenced by Accession Choice and Co-Existing Weeds"

_plants, 2023, doi:10.3390/plants12132401_

Round 1
Reviewer 1 Report
The manuscript investigates the allelopathic activity of three buckwheat accessions against weed germination and growth. Moreover, the research delves deeper by investigating the phenolic profile of the different accessions in crops, both alone and in response to the weed presence. The studies are conducted at three levels: shoots, roots, and root exudates. The authors' proposed experimental design is appropriate, as is the methodology employed to identify and quantify phenols in plant samples.
Some suggestions aimed at improving the manuscript are:
1. Although the title is correct and very realistic, I suggest introducing the term “allelopathy” to attract a greater number of readers, especially when the discussion primarily deals with this topic.
2. The introduction is well-written, but in my opinion, including a paragraph that describes and emphasizes the importance of the allelopathic activity of phenolic compounds could be necessary.
3. Throughout the text, change “grams” by “g”.
4. Is plant weight referred to as fresh or dry weight?
5. The presentation of the results is very clear. However, due to the large number of compounds analyzed and the different accessions and treatments, it is challenging to draw conclusions related to the allelopathic activity of individuals phenols.
6. In the discussion, after some general comments, the authors deal specifically with phenolic changes in different accessions in response to co-cultivation with weeds. I suggest that the authors move the phrase "The findings of this study …….", from lines 525-536, p. 16, to the end of the discussion. The sentence is very interesting and summarizes the results, deserving a more prominent position. Furthermore, this sentence could be expanded by mentioning some of the phenolic compounds that play a more relevant role in allelopathic fighting and whether these activities have been previously reported in other plant species. This would allow for the future use of these natural compounds alone or in combination as allelochemicals.
7. Font size and details of Figures 1 and 2 need to be improved.
Author Response
Reviewer 1
The manuscript investigates the allelopathic activity of three buckwheat accessions against weed germination and growth. Moreover, the research delves deeper by investigating the phenolic profile of the different accessions in crops, both alone and in response to the weed presence. The studies are conducted at three levels: shoots, roots, and root exudates. The authors' proposed experimental design is appropriate, as is the methodology employed to identify and quantify phenols in plant samples.
Some suggestions aimed at improving the manuscript are:
- Although the title is correct and very realistic, I suggest introducing the term “allelopathy” to attract a greater number of readers, especially when the discussion primarily deals with this topic.
According to referee’s suggestion, the term ‘allelochemical’ has been introduced in the title.
- The introduction is well-written, but in my opinion, including a paragraph that describes and emphasizes the importance of the allelopathic activity of phenolic compounds could be necessary.
A new paragraph has been included in the introduction highlighting the importance of phenolic acids in buckwheat and their role in the suppression of weed’s development.
- Throughout the text, change “grams” by “g”.
Grams has been changed by g thorough the text
- Is plant weight referred to as fresh or dry weight?
Plant weight is referred to fresh plant weight. This has been included in the text where appropriate now.
- The presentation of the results is very clear. However, due to the large number of compounds analyzed and the different accessions and treatments, it is challenging to draw conclusions related to the allelopathic activity of individuals phenols.
We totally agree with the referee. Therefore, we have highlighted the potential allelochemical role of individual compounds in some parts of the discussion, but also the importance in this plant interference phenomenon of the plethora of compounds released or accumulated by buckwheat plants.
- In the discussion, after some general comments, the authors deal specifically with phenolic changes in different accessions in response to co-cultivation with weeds. I suggest that the authors move the phrase "The findings of this study …….", from lines 525-536, p. 16, to the end of the discussion. The sentence is very interesting and summarizes the results, deserving a more prominent position. Furthermore, this sentence could be expanded by mentioning some of the phenolic compounds that play a more relevant role in allelopathic fighting and whether these activities have been previously reported in other plant species. This would allow for the future use of these natural compounds alone or in combination as allelochemicals.
The sentence phrase "The findings of this study …….", from lines 525-536, p. 16 has been moved to the end of the discussion as suggested by the referee. Specific mentions to allelochemical properties of individual phenolic compounds have been done along the whole discussion.
- Font size and details of Figures 1 and 2 need to be improved.
We have tried to make the numbers and letters as clear as possible in figures 1 and 2. Unfortunately, the software does not allow us to modify the graphics as much as we would like. We have improved also the legend of the figures to give more information about the different colors of the graphics.
Reviewer 2 Report
The study entitled "Secondary Metabolites Accumulation Pattern in Buckwheat is Strongly influenced by Accession Choice and Co-Existing Weeds "is certainly very interesting for the readers of the journal. The article is original and meaningful for the contemporary scientific literature. I suggest that we accept the work, but I would recommend a few changes to the authors. The experimental section provides enough details on sampling and analytical procedures, the discussion of the results is quite clear and the conclusions are logical.Additionally I suggest to cites this article in the introduction to improve the quality of the work:
1) Parisi, M., Verrillo, M., Luciano, M. A., Caiazzo, G., Quaranta, M., Scognamiglio, F., ... & Fabbrocini, G. (2023). Use of Natural Agents and Agrifood Wastes for the Treatment of Skin Photoaging. Plants, 12(4), 840.
2)Verrillo, M., Koellensperger, G., Puehringer, M., Cozzolino, V., Spaccini, R., & Rampler, E. (2023). Evaluation of Sustainable Recycled Products to Increase the Production of Nutraceutical and Antibacterial Molecules in Basil Plants by a Combined Metabolomic Approach. Plants, 12(3), 513. Furthermore, I suggest to specify the type of standard used for the identification of secondary cIn addition, I would suggest that authors attach a graphic abstract to increase the interest of readers. For these reasons, I suggest that the overall scientific quality of the work is compatible with the standards of the journal so the article should be accepted with minor revisions.
Author Response
Reviewer 2
The study entitled "Secondary Metabolites Accumulation Pattern in Buckwheat is Strongly influenced by Accession Choice and Co-Existing Weeds "is certainly very interesting for the readers of the journal. The article is original and meaningful for the contemporary scientific literature. I suggest that we accept the work, but I would recommend a few changes to the authors. The experimental section provides enough details on sampling and analytical procedures, the discussion of the results is quite clear and the conclusions are logical. Additionally
- I suggest to cite these articles in the introduction to improve the quality of the work: Parisi, M., Verrillo, M., Luciano, M. A., Caiazzo, G., Quaranta, M., Scognamiglio, F., ... & Fabbrocini, G. (2023). Use of Natural Agents and Agrifood Wastes for the Treatment of Skin Photoaging. Plants, 12(4), 840; and Verrillo, M., Koellensperger, G., Puehringer, M., Cozzolino, V., Spaccini, R., & Rampler, E. (2023). Evaluation of Sustainable Recycled Products to Increase the Production of Nutraceutical and Antibacterial Molecules in Basil Plants by a Combined Metabolomic Approach. Plants, 12(3), 513.
The two references have been included in the introduction following referee’s suggestion.
- Furthermore, I suggest to specify the type of standard used for the identification of secondary compounds.
The standards use were the pure compounds of those metabolites identified in the chemical analyses. This has been identified in the material and methods section now.
- In addition, I would suggest that authors attach a graphic abstract to increase the interest of readers.
The graphical abstract has been attached.